# Investigation of the Impact of Environmental Parameters on Breath Frequency Measurement by a Textile Sensor

**DOI:** 10.3390/s20041179

**Published:** 2020-02-21

**Authors:** Ewa Skrzetuska, Jarosław Wojciechowski

**Affiliations:** Faculty of Material Technologies and Textile Design, Institute of Material Science of Textiles and Polymer Composites, 116 Zeromskiego Street, Lodz University of Technology, 90-924 Lodz, Poland; jaroslaw.wojciechowski@p.lodz.pl

**Keywords:** sensors, nanomaterials functionalization, textile actuator, textronic, monitor, human body, carbon nanotubes, screen printing

## Abstract

The aim of this work was to develop sensors that enable the monitoring of respiratory frequencies and will be competitive at a global level in replacing conventional electronic sensors based on rigid and uncomfortable materials. The preliminary work carried out showed the real possibility of creating flat fibrous products containing carbon nanotubes with sensory properties. Bearing in mind the production of a textile deformation sensor, textile materials with high elasticity and deformation reversibility were used in the preliminary studies. The authors assumed that it would be possible to conduct registration associated with the measurement of pneumography continuously in various atmospheric conditions and with varying intensification of human physical activity. The conducted experiment allows us to state that the resistance at the level of 10 kΩ is sufficient to collect results of breathing frequency at rest and after physical effort.

## 1. Introduction

The growing popularity of textronic systems (intelligent clothing) is closely associated with progress in the miniaturization of electronics and the development of textiles and textile technologies. They allow us to, among other things, create new solutions in the field of measuring vital functions, which are an important element of each of the previously mentioned fields. Such systems built into the structure of clothing are intended to fulfill their function without disturbing the comfort of the wearer. This is possible due to the use of sensors. The characteristic properties of some raw materials from which textile products are made include piezoelectric and electrostatic properties and shape memory. Materials using these features are called “intelligent”, and they combine the functions of both the sensor and the activator.

The tasks to be fulfilled by new intelligent textiles are within the sphere of human monitoring, e.g., life activities, or detecting threats in the form of chemical and liquid hazardous substances in liquid and volatile forms. Noninvasive or minimally invasive physiological monitoring devices are of great importance for defense purposes and in applications for athletes. The integration of sensors and biosensors directly into clothing is important for, among other things, the development of health care, threat analysis in the case of uniformed services, and monitoring the level of effort in athletes. Intelligent clothing is primarily the integration of electronics with clothing [1,2].

An example of using worn electronics is the Adidas Micoach Elite system, consisting of a T-shirt equipped with numerous sensors that are connected to a small electronic system located on the player’s back. The data are sent to the training staff on an ongoing basis and contain information such as developed speed, distance traveled, amount of energy used, and pulse. This provides the ability to constantly monitor the physical condition of each player. The measurements taken are aimed at reducing the likelihood of injury during excessive exercise [3].

Another textronic product is the “ZollLifeVest” vest, which aims to protect health by taking defibrillation action when irregularities are detected during analysis of the work of the heart. The vest contains two types of electrodes, one for detection and the other for isolating the gel before the action of electricity. It is also equipped with a system enabling the patient to stop a rescue operation in the event of a mistake [4,5].

Due to the increasingly aging society, the number of chronically ill people is increasing. For this reason, the Chronius project was created, which aimed to monitor patients’ health without requiring hospitalization. A T-shirt was created that can be equipped with different sensors depending on the diagnosed disease. The data collected are sent, for example, to the patient’s mobile, and then to the relevant institution in order to interpret them and adapt the treatment method [3,6].

The scope of requirements for textronic products is very wide. First, they must integrate the accuracy and sensitivity of electronic devices with the flexibility of the textile material. Secondly, they should contain a good power system that can be easily removed before washing, include the option of sending collected information, and, further, should be characterized by very durable connections of all components and resistance to damage during use. By meeting all of these requirements, the spectrum of their use would become enormous. However, this remains a challenge for scientists due to a number of limitations. The biggest problems are power systems, maintenance techniques, and creating a reusable product [4].

For some time, printed electronics have been unrivaled in terms of interest among various industries due to the innovative possibilities resulting from their use. They provide various possibilities for creating solar cells, sensors, and various forms of flexible electronics. Carbon-based printed electronics are projected to complement classic silicon-based electronics [7].

In the chosen field, printed electronics are a combination of well-known or modern printing processes with innovative pastes with conductive properties. This combination allows for printing on flexible substrates such as textiles. Currently, some barriers have been identified associated with the production of sufficiently functional inks and substrates and with the method of protecting finished products [7].

In the present work, the selection of carbon-based nanomaterials for the production of inks with electrical properties was determined by their properties. They are characterized by low resistance, return to their initial form after stretching, resistance to external factors, and good mechanical properties. In addition, they can be brought to the form of a liquid suspension with the desired rheology, the viscosity of which can be adapted to the selected printing technique, which is film printing, allowing us to obtain prints with good resolution [7].

The film printing technique is often used due to the simplicity of the entire printing process, low waste of materials, and low production costs.

The use of printing techniques in creating deformation sensors is a method being increasingly considered for clothing functionalization. Prints made on a knitted fabric using pastes based on carbon nanotubes provide a number of new possibilities. Printed sensors are more convenient to use due to their flexibility, non-binding movements, higher sensitivity, and adequate measurement results [8,9].

Research works conducted at the Institute of Material Science of Textiles and Polymer Composites in Lodz, Poland showed a real possibility of creating flat fiber products with sensory properties containing carbon nanotubes. Undertakings related to the use of various types of carbon and electro conductive polymers for these purposes result from global trends in the field of nanotechnology. Their commercial availability and known and defined unique properties, such as low resistivity, high mechanical resistance, and both biological and chemical sensitivity, lead to a natural need for research in the field of physics and chemistry of polymers and a desire to apply new solutions in everyday life [9].

The aim of this work is to present the outcome of textile sensor analysis of changes in electrical resistance during the rest and physical activity of a human wearing a T-shirt with a printed sensor in changing conditions of humidity and temperature. We decided to use a previously prepared textile actuator with a screen-printed sensor on a T-shirt to measure the electrical resistance of the chest part of the garment [9,10,11,12]. From previous research, it is known that the value of the electrical impedance depends on, among other things, the volume of air in the lungs, the volume and speed of blood flow in the blood vessels, and changes in the shape or displacement of internal organs during breathing. Chest impedance, i.e.,pneumography measurements, is primarily influenced by two components: base impedance, which is assumed to be immutable, and variable impedance, which is dependent on changes in air volume. As a result of inspiration, the volume of air in the chest increases, which results in a decrease in the conductivity of the chest [13,14,15,16]. In addition, when the chest volume increases, the conduction path between the electrodes is extended, which results in an increase in impedance. In their report, Amit K. Gupta presented the relationship between impedance and the amount of air in the chest, proving that this relationship is approximately linear [14].

In order to monitor changes in lung volume, fibrous materials are often used; these are also used to monitor movement, working as strain gauges. The best results are achieved for fibrous structures that combine shape memory (elastomeric fibers) and piezoresistive features. The shape memory of elastomeric fibers is determined by their extremely high reversible deformability. However, elastomeric fibers are, by nature, electro-insulating. They exhibit piezoresistive effects only after preliminary modification providing the fibers with electrically conductive properties.

## 2. Materials and Methods

### 2.1. The Research Aim

The aim of this research was to measure the electrical resistance of an elaborated textile sensor to determine changes in electrical resistance during the rest and physical activity of a human wearing a T-shirt with a printed sensor in changing conditions of humidity and temperature.

The purpose of the work was to produce and analyze a printed respiratory rate sensor. In the scope of this research, the focus was primarily put on analyzing the impact of environmental conditions such as temperature and humidity on the conductive properties of the printouts obtained and checking the effect of printing on the sensory properties of knitted fabric. We hypothesized that after applying the printing composition to textile substrates, various changes in the usable and sensory properties of the product would be noticeable depending on the environmental conditions and physical activity of the user.

### 2.2. The Research Apparatus

This study aimed at checking the impact of mechanical stimulus on changes in the electrical resistance of knitted fabric samples printed with conductive pastes. In addition, it was possible to assess the impact of the raw material on the surface conductive properties of the printing composition.

The measuring apparatus in the selection phase of the optimal variant was an INSTRON testing machine from the 5900 series, a Keithley model 2000 digital multimeter, and a computer equipped with software registering resistance changes during mechanical deformation of the knitted fabrics. The samples were cut into 250 × 50 mm strips, then manually conductive thread, to which the electrodes of the multimeter were connected, was sewn into them. The deformation took place in ten cycles of tension and annealing of the knitted fabric samples for a length of 20 mm. Based on the obtained results, the textile substrate with the most repeatable sensory signal was selected.

Then, a Keithley model 2000 digital multimeter and a computer equipped with software recording changes in the T-shirt’s resistance via the printed sensors during breathing were placed in a large WEISS chamber.

The large air-conditioned chamber provided measurement conditions from −20 °C to +50 °C and from 5% Rh to 90% Rh.

### 2.3. The Research Material

One printing composition was used in this work. It was an aqueous dispersion of carbon nanotubes with the trade name AQUACYL™ AQ0101 from Nanocyl. This dispersion contained 1% Multi-Wall Carbon Nanotube Aqueous Dispersion, DI water, and surfactant from line NC7000™ [17]. The printing paste had a viscosity of 47 Pa*s. The thickness of the applied layer was 13µm.The print was made with the use of a screen-printing technique on the knitwear. The knitted fabrics were printed using anMS-300FRO screen printing machine. This is a multifunctional device that allows printing on various types of surfaces.

One aspect of the research was the selection of the most optimal sensor shape for pneumography measurements. For this purpose, printouts were made, with the help of which textile surfaces were modified and electrically conductive paths in various shapes and sizes were obtained (Figure 1). Then, tests were conducted under conditions simulated with the Instron testing machine.

Bearing in mind the production of a textile deformation sensor, textile materials with high elasticity and deformation reversibility were used as the basis. The textile substrates were selected based on preliminary studies that assessed their effectiveness for athletic shirts. The selected fabric consisted of 85% polyester yarn (PES) and 15% polyurethane yarn (PU). The surface mass of the fabric was 185 g/m^2^, and its thickness was 0.45 mm.

After preliminary tests using a testing machine that simulated the inspiration and exhalation process, it was determined that the most optimal variant of printing was full filling at a width of 6 cm. The variant of the printed square gave the most repeatable and uninterrupted signal at 20% unidirectional deformation (Figure 2).

Before starting to make the T-shirt, the unit pressure exerted by the tested knitwear was determined. Based on the results of the endurance testing of knitted fabric samples and the taken dimensions of models of people, the pressure exerted by the fabric was calculated in accordance with La Place’s relationship and the dimensions of the product in the free state were determined. Knowing the results of these calculations, it was possible to properly cut and make the T-shirt. In the finished T-shirt, in places where the edges of the print were, a conductive thread was sewn, leading it inside the side seams to the bottom of the T-shirt.

The height of the print was 6cm over the entire width of the T-shirt, terminated with conductive thread sewn on the sides, which is illustrated in the technical drawing shown in Figure 3.

The thermal insulation value of the fabric was 0.06 m^2^K/W. The T-shirt had good air permeability, thanks to which the moisture transfer from the body to the outside was correct. The design of the jersey allowed a proper heat balance to be maintained.

The drawing uses an original photo of the T-shirt used in the experiment. There were sensors under the clothing that measured temperature and humidity. The T-shirt was fitted and tight on the subject, a young person (ca. 24 years). Conductive threads ran along the seam and were connected to a multimeter.

### 2.4. The Course of the Research

The measurements were made in a large-size chamber enabling the control of climatic conditions, located in the Institute of Material Science of Textiles and Polymer Composites at the Lodz University of Technology, Poland. The first stage of the experiment was to fit a T-shirt containing sensors to the subject’s body so that it adhered properly to it, resulting in more reliable results. The tests were carried out at varying temperature and humidity in the following proportions:temperature 20 °C, humidity 50%;temperature 25 °C, humidity 60%;temperature 25 °C, humidity 90%;temperature 30 °C, humidity 90%.

The study was conducted for four days on two women of 24 years of age and of a similar height and weight. Five repeats of measurements were performed for the same ambient conditions in the climatic chamber and for each of the two test participants. Every day the persons conducted tests in a variant of conditions, i.e., 20/50 on the first day, 25/60 on the second day, etc.

The rest of the article presents the averages calculated from the measurements of both women, which are also presented in the example charts.

The experiment was conducted while the subject was at rest and after a five-minute run. All reactions were recorded by a computer program connected to a Keithley multimeter. The obtained results were used to determine the change in resistance over time.

## 3. Results

### 3.1. Physical Parameter Measurement

Table 1 presents a summary of the subject’s temperature and humidity values at rest and after physical activity under the abovementioned conditions. The air flow towards the participants was constant for all air conditioning conditions and was 0.4 m/s.

When analyzing the results contained in the table, it can be observed that the participants taking part in the experiment showed decreased body temperature for climatic conditions close to a normal climate. After physical activity, it was observed in all cases that physical exertion caused an increase in humidity under clothing, which is associated with the secretion of sweat. In all cases, the underwater temperature increased with physical exertion.

The material from which the T-shirts were made, i.e., PES/PU fabric printed with an ink composition based on carbon nanotubes, was subjected to utilization processes such as friction and a 25-fold washing cycle. After the washing process, the resistance deteriorated to about 8.7 × 10^5^ Ω, and after 1500 cycles of friction (the rubbing medium was pure PES/PU knitwear), the resistance was at the level of about 3.6 ×10^5^ Ω.

### 3.2. Experimental Investigation of Changes in Resistance at Rest

The graph in Figure 4 presents a summary of results for all the conditions used in the experiment under which tests were carried out during a five-minute rest. From the graph it can be read that change in humidity had a greater impact than the temperature. We notice changes between the charts for 25/60 and 25/90 in which the temperature factor was the same, while the increase in humidity changed the decrease in electric resistance. Temperature and humidity had a slightly noticeable effect on resistance with a variation coefficient of up to 3%, which we see in Table 2.

### 3.3. Experimental Investigation of Changes in Resistance after Physical Activity

The graph in Figure 5 presents a summary of the results for all conditions used in the experiment under which tests were carried out on the subject after a five-minute physical effort. The temperature and humidity exhibited noticeable effects on the resistance with a variation coefficient of up to 3%, which we see in Table 3. For some hard-to-read small parts of the chart measurements, future work on differential elements of the digital embedded system may come in handy to detect changes in value that are hardly visible on the graph or that can be dismissed as artefacts.

Table 4 shows that the number of breaths increased because after physical activity the human body needs more oxygen.

It is commonly known that with increasing temperature, the resistance of a conductor increases as well. We suspect a similar phenomenon in the results of the measurements. An increase in ambient temperature of 5 °C caused a noticeable increase in the resistance of the textile sensor.

Under normal climate, the resistance was at 8.72 × 10^4^; a temperature rise of 5 °C and 10% Rh caused an increase in resistance of ca. 2.5 kΩ. After another increase of 5 °C, the resistance remained at a similar level. This could be influenced by the increase in humidity under clothing from 40% Rh to 46% Rh, which could compensate the resistance increase (because of the temperature at measure variant 30/60), keeping it at a similar level.

In the case of physical activity, the body temperature of the participants and humidity under clothing increased, which could lead to moisture on the sensors. The resistance in all thermal conditions remained at a similar level.

## 4. Discussion

The aim of this research was to count the average number of breaths by means of measurement of the electrical resistance of the elaborated textile sensor. Changes in electrical resistance during the rest and physical activity of a human wearing a T-shirt with a printed sensor in changing conditions of humidity and temperature corresponded to the number of breaths. The experiment was successful, which can be observed in the graphs obtained on the basis of data collected by the textile sensor. The graphs present the frequency of breaths by the subject after activity and at rest. The average frequency of breaths of the examined person was about 16 breaths per 1 min at rest, and after 5 min of physical activity it was about 23 breaths per 1 min. Expected values are within 14–20 breaths per minute for a healthy adult under normal climate conditions at rest. The difference between the values obtained at rest and after 5 min of physical activity is natural, because after physical activity the human body needs more oxygen, as evidenced by higher frequency of breaths recorded on specific charts.

It was observed that an increase in temperature in the large chamber and increase in humidity caused an increased respiratory rate in the examined person. The result of manually counting the number of breaths coincided with the number of appearing peaks recorded with a multimeter, obtained from sensors printed on the T-shirt. The conducted research has highlighted the fact that when designing clothing products that are to register the number of breaths, one should take into account the changing climatic conditions in which the product will be used. These conditions affect the behavior of fibers in the structure of textile products, e.g., fiber swelling, which is important when registering resistance from sensors on their surface. Based on the results obtained, it can be concluded that breath sensors should be protected/insulated against the effects of external conditions and sweat, or if this is not possible, tests should be carried out in different climatic conditions and their results should be taken into account when calibrating printed sensors, assuming variability in conductivity ranges.

The elaborated T-shirt sensor also requires calibration. For future work, one needs a microcomputer system to monitor the parameters without external laboratory equipment.

## Figures and Tables

**Figure 1 sensors-20-01179-f001:**
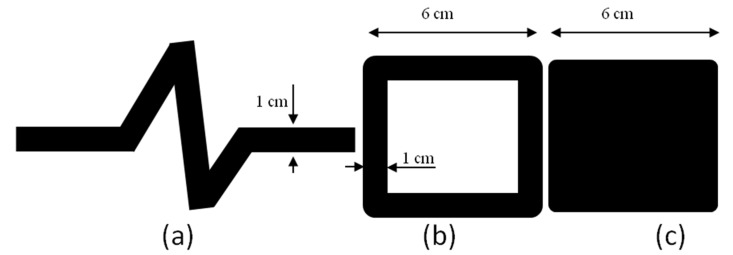
Patterns of printouts: (**a**) curve, (**b**) frame, (**c**) full square.

**Figure 2 sensors-20-01179-f002:**
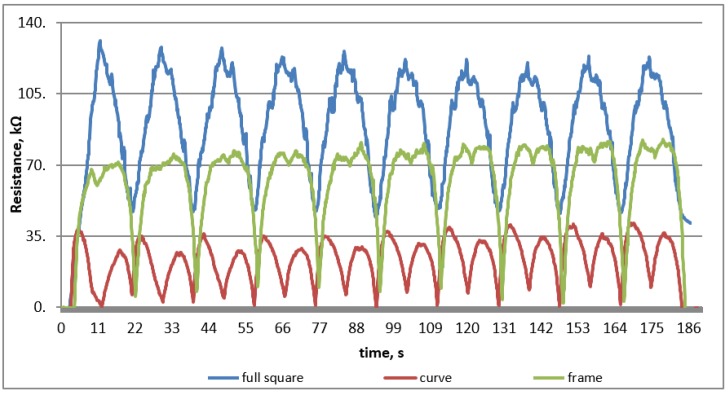
The course of resistance changes during the sensory sensitivity test.

**Figure 3 sensors-20-01179-f003:**
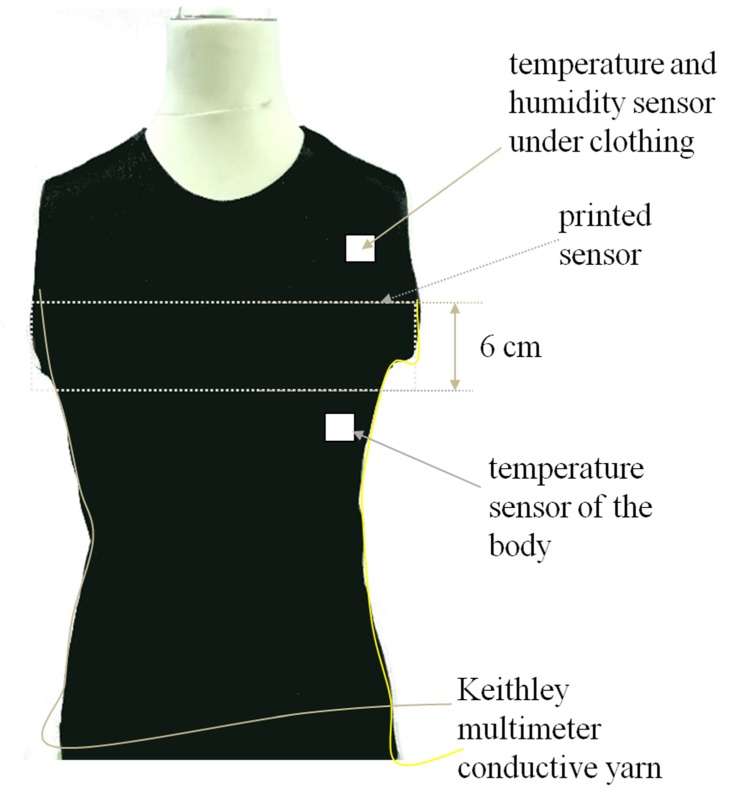
T-shirt with imprinted sensor.

**Figure 4 sensors-20-01179-f004:**
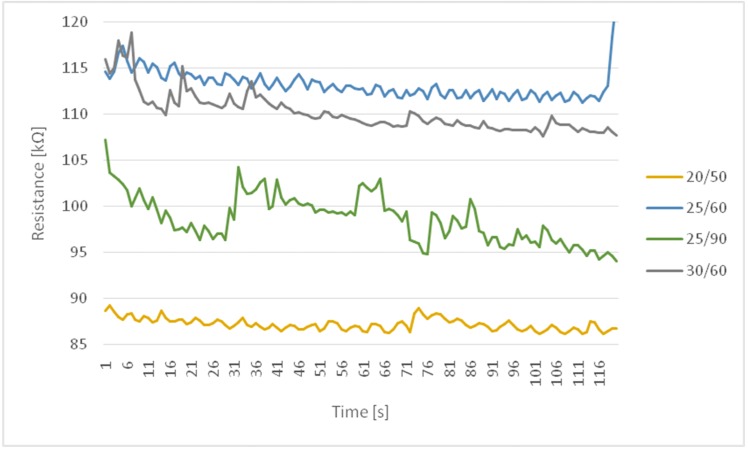
Summary of changes in resistance at rest.

**Figure 5 sensors-20-01179-f005:**
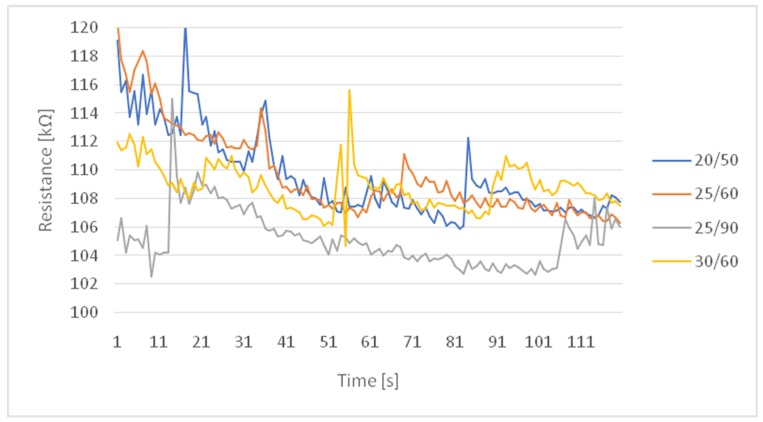
Summary of changes in resistance after motion.

**Table 1 sensors-20-01179-t001:** Temperature and humidity values at rest and after physical activity.

Temperature [°C]/ Humidity [%] (Chamber Settings)	Physical Parameters (at Human Body)	Rest (5 min)	Physical Activity (after 5 min)
20/50	Body temperature [°C]	35.6	34.7
	Temp under clothing [°C]	30.5	31.4
	Ambient temperature [°C]	36	36
	Moisture under clothing [%]	35	35
	Ambient humidity [%]	41	37
25/60	Body temperature [°C]	35.4	34.5
	Temp under clothing [°C]	33.3	33.9
	Ambient temperature [°C]	25	25
	Moisture under clothing [%]	40	42
	Ambient humidity [%]	55	54
25/90	Body temperature [°C]	35.9	35.9
	Temp under clothing [°C]	34.6	34.7
	Ambient temperature [°C]	25	25
	Moisture under clothing [%]	48	52
	Ambient humidity [%]	70	70
30/60	Body temperature [°C]	35.7	36
	Temp under clothing [°C]	34.1	35.5
	Ambient temperature [°C]	30	30
	Moisture under clothing [%]	46	51
	Ambient humidity [%]	60	61

**Table 2 sensors-20-01179-t002:** Statistical data and variation coefficients at rest.

Measure Variant	Average Resistance	Rest (5 min)	
		Standard Deviation	Coefficient of Variation
20/50	8.72 × 10^4^	6.49 × 10^2^	1%
25/60	1.13 × 10^5^	1.58 × 10^3^	1%
25/90	9.86 × 10^4^	2.61 × 10^3^	3%
30/60	1.10 × 10^5^	2.15 × 10^3^	2%

**Table 3 sensors-20-01179-t003:** Statistical data and variation coefficients after activity.

Measure Variant	Average Resistance	Physical Activity (5 min)	
		Standard Deviation	Coefficient of Variation
20/50	1.10 × 10^5^	3.06 × 10^3^	3%
25/60	1.10 × 10^5^	3.12 × 10^3^	3%
25/90	1.05 × 10^5^	1.97 × 10^3^	2%
30/60	1.09 × 10^5^	1.61 × 10^3^	1%

**Table 4 sensors-20-01179-t004:** Average number of breaths per minute.

Measure Variant	Number of Breaths	
	Rest (5 min)	Physical Activity (5 min)
20/50	14	20
25/60	16	23
25/90	17	23
30/60	18	26

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
