# Peer review of "Investigation of the Impact of Environmental Parameters on Breath Frequency Measurement by a Textile Sensor"

_sensors, 2020, doi:10.3390/s20041179_

Round 1

Reviewer 1 Report

Aim of the manuscript seems to be interesting but need more understanding and experiments to conclude something.

1.       The authors should follow author’s guideline, when formatting the manuscript. References should be written in chronological order in the text.

2.       Instead of having such a long title for the manuscript, authors can think of something crispy and catchy for the manuscript (just a suggestion)

3.       Abstract can be elaborated a little bit more and try to shorten the length of sentences.

4.       Authors have mentioned so many advantages and one disadvantages concerning the use of screen-printing on silk, what about the resolution of printing? Is it an advantage or disadvantage?

When printing electronic circuits on textiles, the printer should have either good resolution or the circuit should design in such a way that the printing resolution should not matter to the performance of circuit.

5.       Need some English correction for eg., in line 229, …within 60 first seconds…

6.       Do Authors have any schematic or photographs to show their CNT sensor?

7.       How many samples the authors have tested? Is the results varying with people?

8.       Figure 6 is missing

9.       Since authors have summarized Figure 3-6 in Figure 7 and is conveying the message, it is not necessary to keep figure 3-6.

10.   The statement authors have made in line 256, 257….’ same, while the increase of humidity changed the increase of electric resistance’ is contradicting the results presented in Figure 7. Increase of humidity change from 25/60 to 25/90, slightly decreased the resistance. For former, resistance is around 1.15*105 and for later, it is around 1.00*105. Moreover, change in resistance is not following any pattern with change in humidity. From Rh 50 to 60, the resistance increases but from 60 to 90, resistance decreases.

11.   Also the breadth count given for the four (different conditions) rest positions are somewhat random. There are some long flat regions, how do you calculate the number of peaks in these regions? For eg., in figure 9, the region between 11 and 21 sec, how many peaks are there?

12.   How do you do all these experiments? Before applying any new environmental condition to the sample (the person under investigation), do you take break or you continuously change the conditions? Does this affect the results? Because your sample of interest may not be same in each case. In reality there are so many things affecting the breadth of physical activity. For eg., the wind.

13.   Figure 12 is wrongly numbered. Again, since the authors have summarized Figure 8-11 in figure 12, figure 8-11 can be removed.

14.   The number of breadths at rest and after physical activity is quite random. There is no specific order in which the breadth count increases. Do you have any explanation for this? How do you correlate this with the resistance change? Even the change in resistance is random with temperature and humidity.

15.   As far as I understood the authors have done the experiments on a single person and concluded the results, they should do it on many people and then conclude the results.

Author Response

Response to Reviewer 1 Comments

At the beginning, we wanted to thank you for your thorough review and valuable comments.

Point 1: The authors should follow author’s guideline, when formatting the manuscript. References should be written in chronological order in the text. 

 Response 1:

As suggested by the reviewer, the authors formatted text according to the magazine's guidelines and improved the references and ordered them chronologically.

Point 2: Instead of having such a long title for the manuscript, authors can think of something crispy and catchy for the manuscript (just a suggestion)

Response 2:

As suggested by the reviewer, the authors corrected the manuscript title.

Point 3: Abstract can be elaborated a little bit more and try to shorten the length of sentences.

Response 3: As suggested by the reviewer, the authors improved the abstract of the manuscript.

Point 4: Authors have mentioned so many advantages and one disadvantages concerning the use of screen-printing on silk, what about the resolution of printing? Is it an advantage or disadvantage?

When printing electronic circuits on textiles, the printer should have either good resolution or the circuit should design in such a way that the printing resolution should not matter to the performance of circuit.

Response 4:

The authors made corrections to the description of printing techniques. We agree with the reviewer that printed electrical circuits on textiles should have good resolution. In the case of textiles, we can modify the parameters of the substrate by applying appropriate finishes, we can also modify the parameters of printing pastes (viscosity, surface tension) so as to obtain sharp contours. Screen printing is thick-layer printing, therefore it has many advantages when printing on textiles with high surface unevenness.

Point 5: Need some English correction for eg., in line 229, …within 60 first seconds

Response 5: The authors submitted the article for language correction

Point 6: Do Authors have any schematic or photographs to show their CNT sensor?

Response 6:

The authors in the revised chapter “2.3. The research material” gave examples of sensors.

Point 7: How many samples the authors have tested? Is the results varying with people?

Response 7: The study was conducted for four days on two women of 24 years of age with similar height and weight. Every day, T-shirt users conducted tests in different air-conditioning conditions.

Five repeats of measurements were performed for the same ambient conditions in the climatic chamber and for each of the two test participants. The rest of the article presents the averages calculated from the measurements of both women and are presented in the example charts. No changes were observed among the participants. The information was introduced in the article in part 2.4.

Point 8: Figure 6 is missing

Response 8: Figure 6 was removed as suggested by the reviewer.

Point 9: Since authors have summarized Figure 3-6 in Figure 7 and is conveying the message, it is not necessary to keep figure 3-6.

Response 9: Figure 3-6 was removed as suggested by the reviewer.

Point 10: The statement authors have made in line 256, 257….’ same, while the increase of humidity changed the increase of electric resistance’ is contradicting the results presented in Figure 7. Increase of humidity change from 25/60 to 25/90, slightly decreased the resistance. For former, resistance is around 1.15*105 and for later, it is around 1.00*105. Moreover, change in resistance is not following any pattern with change in humidity. From Rh 50 to 60, the resistance increases but from 60 to 90, resistance decreases.

Response 10: It is commonly known that with the temperature increase, the resistance of the conductor increases as well. We suspect that a similar phenomenon was observed as the results of the measurements. An increase in ambient temperature of 5 °C causes a noticeable increase in the resistance of the textile sensor.

   Under normal climate, the resistance is at 8.72 *104, a temperature rise of 5°C and 10% Rh causes an increase in resistance by an order of magnitude. Another increase of 5°C shows that the resistance remains at a similar level. This can be influenced by an increase in humidity under clothing from 40% Rh to 46% Rh, which could compensate the resistance increase (because of temperature at measure variant 30/60) keeping it at a similar level.

   In the case of physical activity, the body temperature of the participants and humidity under clothing increased, which could lead to moisture of the sensors. The resistance in all thermal conditions remained at a similar level. The information was introduced in the article in part 3.3.

Point 11: .   Also the breadth count given for the four (different conditions) rest positions are somewhat random. There are some long flat regions, how do you calculate the number of peaks in these regions? For eg., in figure 9, the region between 11 and 21 sec, how many peaks are there?

Response 11: The graph presents a summary of results for all conditions used in the experiment under which tests were carried out on the subject after a 5-minute physical effort. The temperature and humidity exhibit noticeable effect on resistances with coefficient variation up to 3% what we see at Table 3. For some hard-to-read small parts of the charts measurements, a future work of differential element of digital embedded system can come in handy that can detect a change in value that is hardly visible on the graph or it can be bypass as an artefact. The information was introduced in the article in part 3.3.

Point 12: How do you do all these experiments? Before applying any new environmental condition to the sample (the person under investigation), do you take break or you continuously change the conditions? Does this affect the results? Because your sample of interest may not be same in each case. In reality there are so many things affecting the breadth of physical activity. For eg., the wind.

Response 12: The air flow towards the participants was constant for all air conditioning conditions and was 0.4 m/s. The study was conducted for four days on two women of 24 years . Every day, T-shirt users conducted tests in different air-conditioning conditions. Women took 30 minutes break between measurements.

Point 13: Figure 12 is wrongly numbered. Again, since the authors have summarized Figure 8-11 in figure 12, figure 8-11 can be removed.

Response 13: Figure 8-11 was removed as suggested by the reviewer.

Point 14: The number of breadths at rest and after physical activity is quite random. There is no specific order in which the breadth count increases. Do you have any explanation for this? How do you correlate this with the resistance change? Even the change in resistance is random with temperature and humidity.

Response 14:

Observing the experiment participants, we noticed that the number of breaths per minute increases with increasing temperature and physical effort. Further work will certainly need to take into account the sweating of people and the fact that multidirectional stretching occurs in clothing. First of all, this multidirectional stretching associated with, among others, the movement of the hands can affect the occurring artifacts and interference in measurements.

Point 15: As far as I understood the authors have done the experiments on a single person and concluded the results, they should do it on many people and then conclude the results.

Response 15: The study was conducted on two women of 24 years. The experiment should in the future be extended to people of different sexes and ages. Unfortunately, this involves expensive volunteer insurance.

Reviewer 2 Report

The manuscript entitled "Testing of deformation sensor printed whit carbon nanotubes on T-shirt for electrical resistance parameter in changing humidity and temperature conditions during rest and physical activity of human" presented by Ewa Skrzetuska and JarosÅ‚aw Wojciechowski describes measurements of the resistance of CNTs printed on a T-shirt as an indication of breathing activity. The authors find with no surprise that it is possible to measure a breathing rate and to distinguish between physical activity and resting. Also not surprising is the fact that the results depend on the humidity and the temperature. The authors missed to clarify if other environmental changes will have any impact. E.g. it is well known that the resistance of carbon nonmaterial changes with the air quality. NOx has a tremendous effect on the resistance. Furthermore it remains unclear if such a T-Shirt will loose its function when it becomes washed. What is the life-cycle of such an "intelligent" textile? This few questions show that there are many aspects which could affect the findings of this study. In general I do not think that this manuscript fits the aims and the scope of a chemical sensor Journal like MDPI sensors. I do not see any wider impact. The presentation of this manuscript does also not match to those articles usually published by MDPI Sensors. The results section is a list of graphs with almost not interpretation. The discussion section does not go in detail. It remains unclear why the introduction explains the body's heat balance when this is not used further on in the manuscript. All together my recommendation is to reject this submission.

The following issues might be helpful in preparing another version of this manuscript:

Abstract: The abstract seems to be extremely short and does consist of the aim of this study only. Usually abstracts should attract readers when searching for publications. Therefore I suggest to extent the abstract by summarizing the most important results.  Introduction reads like an article on Wikipedia and not for a scientific journal. Table 1: Remains unclear why a table is used when only one entry is displayed. This 4 numbers can also be presented in text form. Even when statistics are displayed in Table 3 and Table 4, it is not clear how reproducible these data are. It seems that only a single measurement was performed for each condition (e.g. 20° / 50% humidity) 

Author Response

Response to Reviewer 2 Comments

At the beginning, we wanted to thank you for your thorough review and valuable comments.

Point 1: The authors missed to clarify if other environmental changes will have any impact.

Response 1:In the case of physical activity, the body temperature of the participants and humidity under clothing increased, which could lead to moisture of the sensors. The resistance in all thermal conditions remained at a similar level. The information was introduced in the article in part 3.3. We have not tested whether the quality of air has any impact on the textile sensor. We only controlled the climatic conditions.

Point 2: It remains unclear if such a T-Shirt will loose its function when it becomes washed. What is the life-cycle of such an "intelligent" textile? 

Response 2: The material from which the T-shirts were made, i.e. PES / PU fabric printed with an ink composition based on carbon nanotubes, was subjected to utilization processes such as friction and a 25-fold washing cycle. After the washing process, the conductivity deteriorated to about 8.7 * 105 Ω and after 1500 cycles of friction (the rubbing medium was pure PES / PU knitwear) the conductivity was at the level of about 3.6 * 105 Ω.

Point 3: The results section is a list of graphs with almost not interpretation. The discussion section does not go in detail. It remains unclear why the introduction explains the body's heat balance when this is not used further on in the manuscript.

Response 3: It has been removed individual graphs from the article, leaving collective figures. Interpretations of test results have been added under charts and tables.

Point 4: It remains unclear why the introduction explains the body's heat balance when this is not used further on in the manuscript. 

Response 4: We agree with reviewer and we corrected the introduction and removed the paragraph about the body’s heat balance topic.

Point 5: The abstract seems to be extremely short and does consist of the aim of this study only. Usually abstracts should attract readers when searching for publications. Therefore I suggest to extent the abstract by summarizing the most important results.  Introduction reads like an article on Wikipedia and not for a scientific journal.

Response 5: As suggested by the reviewer, the authors improved abstract and made major changes in the introduction.

Point 6:  Table 1: Remains unclear why a table is used when only one entry is displayed. This 4 numbers can also be presented in text form.

Response 6: The authors deleted Table 1, replacing it with a word description.

Point 7:Even when statistics are displayed in Table 3 and Table 4, it is not clear how reproducible these data are. It seems that only a single measurement was performed for each condition (e.g. 20° / 50% humidity)  

Response 7: The study was conducted for four days on two women of 24 years of age with similar height and weight. Five repeats of measurements were performed for the same ambient conditions in the climatic chamber and for each of the two test participants. Every day the persons conducted tests in each variant of conditions i.e. 20/50 in first day, 25/60 in second day etc. The air flow towards the participants was constant for all air conditioning conditions and was 0.4 m/s. Women took 30 minutes break between measurements. The rest of the article presents the averages calculated from the measurements of both women and are presented in the example charts. No changes of resistance were observed among the participants. The information was introduced in the article in part 2.4.”The course of the Research”.

Further work will certainly need to take into account the sweating of people and the fact that multidirectional stretching occurs in clothing. First of all, this multidirectional stretching associated with, among others, the movement of the hands can affect the occurring artifacts and interference in measurements.

The experiment should in the future be extended to people of different sexes and ages. Unfortunately, this involves expensive volunteer insurance.

Round 2

Reviewer 1 Report

Authors have answered most of the questions raised and made a considerable improvement in the revised manuscript, which is appreciable.

There are some more things, which need more clarity:

In line 239,   authors mentioned ‘After the washing process, the conductivity deteriorated to about 8.7 * 105 Ω and after…’, unit of the value says it is not conductivity, it is the resistance.

Similar thing in the next line also.

In line 243, authors have given a sub heading, which sounds more like a caption to Figure 4. Authors can write a different sub heading like ‘Experimental investigation of change in resistance at rest’ (this is just a suggestion, you can write according to your wish). Authors have mentioned in lines 249-251 that resistance increases with increase in humidity for same temperature. But from figure 4 and Table 2, it looks like resistance decreased slightly from 1.13*105 Ω (or 11.3*104 Ω) to 9.86*104 Ω when humidity changed from 60 % to 90 %. Please clarify. In line 253, authors have given a sub heading, which sounds more like a caption to Figure 5. Line 255 needs editing In lines 277-279, authors have mentioned that ‘Under normal climate, the resistance is at 8.72 *104, a temperature rise of 5°C and 10% Rh causes an increase in resistance by an order of magnitude. Another increase of 5°C shows that the resistance remains at a similar level’. Which is not completely true. If you see the Table 2, the average resistance for 20/50 is 8.72*104 Ω and that at 25/60 is 1.13*105 Ω (11.3*104 Ω), which is not an order of difference, in fact this is not a huge difference.

Author Response

Response to Reviewer 1 Comments

Point1:

In line 239,   authors mentioned ‘After the washing process, the conductivity deteriorated to about 8.7 * 105 Ω and after…’, unit of the value says it is not conductivity, it is the resistance.

Similar thing in the next line also.

Response 1: 

As suggested by the reviewer, the authors corrected the conductivity to resistance

Point 2:

In line 243, authors have given a sub heading, which sounds more like a caption to Figure 4. Authors can write a different sub heading like ‘Experimental investigation of change in resistance at rest’ (this is just a suggestion, you can write according to your wish)

Response 2: 

As suggested by the reviewer, the authors changed sub heading to: Experimental investigation of change in resistance at rest’

Point 3:

Authors have mentioned in lines 249-251 that resistance increases with increase in humidity for same temperature. But from figure 4 and Table 2, it looks like resistance decreased slightly from 1.13*105 â„¦ (or 11.3*104 â„¦) to 9.86*104 â„¦ when humidity changed from 60 % to 90 %. Please clarify.

Response 3:

As suggested by the reviewer, the authors changed  the increase to decrease: "while the increase of humidity changed the decrease of electric resistance"

Point 4:

In line 253, authors have given a sub heading, which sounds more like a caption to Figure 5.

Response 4:

As suggested by the reviewer, the authors changed sub heading to:  Experimental investigation of change in resistance after physical activity

Point 5:

Line 255 needs editing 

Response 5:

Line was removed.

Point 6:

In lines 277-279, authors have mentioned that ‘Under normal climate, the resistance is at 8.72 *104, a temperature rise of 5°C and 10% Rh causes an increase in resistance by an order of magnitude. Another increase of 5°C shows that the resistance remains at a similar level’. Which is not completely true. If you see the Table 2, the average resistance for 20/50 is 8.72*104 â„¦ and that at 25/60 is 1.13*105 â„¦ (11.3*104 â„¦), which is not an order of difference, in fact this is not a huge difference.

Response 6:

As suggested by the reviewer, the authors changed "order of magnitude" to a value:

"Under normal climate, the resistance is at 8.72 *104, a temperature rise of 5°C and 10% Rh causes an increase in resistance by ca.2.5 kΩ. "

Reviewer 2 Report

The authors revised the manuscript in accordance to the reviewers suggestions. Therefore I recommend to accept the manuscript as it is.

Author Response

We wanted to thank you for your thorough review and valuable comments and inform that we will use the English Editing Services of MDPI for language corrections.